# Population genomics reveals the origin and asexual evolution of human infective trypanosomes

William Weir[1], Paul Capewell[1], Bernardo Foth[2], Caroline Clucas[1], Andrew Pountain[1], Pieter Steketee[1], Nicola Veitch[3], Mathurin Koffi[4†], Thierry De Meeûs[5,6], Jacques Kaboré[6,7†], Mamadou Camara[8†], Anneli Cooper[1], Andy Tait[1], Vincent Jamonneau[5,6†], Bruno Bucheton[5,8†], Matt Berriman[2], Annette MacLeod[1*†]

[1]Wellcome Centre for Molecular Parasitology, College of Medical, Veterinary and Life Sciences, Institute of Biodiversity Animal Health and Comparative Medicine, University of Glasgow, Glasgow, United Kingdom; [2]Sanger Institute, Wellcome Trust Genome Campus, Cambridge, United Kingdom; [3]West Medical Building, Office 350, University of Glasgow, Glasgow, Scotland; [4]UFR Environnement, Laboratoire des Interactions Hôte-Microorganismes-Environnement et Evolution (LIHME), Université Jean Lorougnon GUEDE, Daloa, Côte d'Ivoire; [5]Institut de Recherche pour le Développement, Campus International de Baillarguet, Montpellier, France; [6]Centre International de Recherche-Développement de l'Elevage en zone Subhumide, Bobo-Dioulasso, Burkina Faso; [7]Université Polytechnique de Bobo-Dioulasso, UFR Sciences et Techniques, Bobo-Dioulasso, Burkina Faso; [8]Programme National de Lutte contre la Trypanosomiase Humaine Africaine, Conakry, Guinea

**\*For correspondence:** annette. macleod@glasgow.ac.uk

[†]Member of TrypanoGEN, part of the H3Africa consortium

**Competing interests:** The authors declare that no competing interests exist.

**Abstract** Evolutionary theory predicts that the lack of recombination and chromosomal re-assortment in strictly asexual organisms results in homologous chromosomes irreversibly accumulating mutations and thus evolving independently of each other, a phenomenon termed the Meselson effect. We apply a population genomics approach to examine this effect in an important human pathogen, *Trypanosoma brucei gambiense*. We determine that *T.b. gambiense* is evolving strictly asexually and is derived from a single progenitor, which emerged within the last 10,000 years. We demonstrate the Meselson effect for the first time at the genome-wide level in any organism and show large regions of loss of heterozygosity, which we hypothesise to be a short-term compensatory mechanism for counteracting deleterious mutations. Our study sheds new light on the genomic and evolutionary consequences of strict asexuality, which this pathogen uses as it exploits a new biological niche, the human population.

## Introduction

Obligate asexual reproduction has been argued to carry considerable negative evolutionary consequences (*Maynard Smith, 1986*) hence most clonal species undergo some degree of recombination, albeit infrequent, which generates novel genotypes (*Heitman, 2006*). Here we investigate the reproductive strategy of a putatively asexual yet successful human pathogen, *Trypanosoma brucei gambiense* Group 1, the main causative agent of human African trypanosomiasis, which contrasts with closely-related, sexually reproducing sub-species. *T.b. gambiense* has been divided into two groups.

**eLife digest** An organism's genetic 'blueprint' is encoded in DNA packaged within structures called chromosomes. Most organisms have two copies of each chromosome and, through sexual reproduction, the DNA within a pair of chromosomes can recombine randomly in a process that could be likened to shuffling a deck of cards. This process generates genetic diversity and means that any undesirable combinations of genes or mutations can be eliminated from the population by natural selection. While these activities are thought to promote the long-term survival of a species, some organisms appear not to have sex at all.

Evolutionary theory predicts that 'asexual' organisms should face extinction in the long-term and that a lack of sexual recombination should leave a characteristic genetic 'signature' in their DNA. The theory also predicts that pairs of chromosomes will evolve independently, a phenomenon that is termed the 'Meselson effect'. However, while it was first predicted almost twenty years ago, evidence for this signature has been elusive.

Now, Weir et al. have used the asexual parasite (*Trypanosoma brucei gambiense*), which causes African sleeping sickness in humans, to search for signs of the Meselson effect. Sequencing the whole genome of a large number of parasites revealed that the population of this parasite arose from a single individual within the last 10,000 years. Over this time, mutations have built up and the lack of sexual recombination means that the two chromosomes in each pair have evolved independently of the other. These results provide the first demonstration of the Meselson effect at a genome-wide level in any organism.

Weir et al. also uncovered evidence that this parasite uses a mechanism called "gene conversion" to compensate for its lack of sex. This mechanism essentially repairs the inferior, or mutated, copy of a gene on a chromosome by 'copying and pasting' the superior copy from the chromosome's partner. The findings also suggest that gene conversion can only go some way to compensating for a lack of sex. A future challenge will be to investigate how effective this mechanism can be in the long term and to predict whether the parasite will ultimately become extinct.

*T.b. gambiense* Group 1, found in West and Central Africa, is the main human-infective sub-species, causing >97% of all human cases of trypanosomiasis (*Simarro et al., 2010*). *T.b. gambiense* Group 2 was detected in the 1980/90s in Côte d'Ivoire and Burkina Faso but may now be extinct (*Capewell et al., 2013*). A third human infective sub-species, *T.b. rhodesiense,* is found in East Africa and causes <3% of human cases (*Simarro et al., 2010*). Each of these human infective sub-species appears to have arisen independently from the non-human infective *T. brucei* and possesses a different mechanism of human infectivity (*Capewell et al., 2013*; *Capewell et al., 2011*; *Uzureau et al., 2013*; *Van Xong et al., 1998*). All sub-species of *T. brucei*, with the important exception of *T.b. gambiense* Group 1, show evidence for mating in natural populations (*Capewell et al., 2013*; *Duffy et al., 2013*; *Gibson and Stevens, 1999*) and have the ability to undergo sexual reproduction with Mendelian allelic segregation and independent assortment of unlinked markers (*Cooper et al., 2008*; *MacLeod et al., 2005*). In addition, haploid gametes have been observed in *T.b. brucei* (*Peacock et al., 2014*) and although meiosis genes appear to be expressed in *T.b. gambiense* Group 1 (*Peacock et al., 2014*), no haploid gametes have ever been observed in these parasites (*Peacock et al., 2014*). This is consistent with clonality in all Group 1 populations analysed (*Koffi et al., 2009*; *Morrison et al., 2008*; *Tibayrenc and Ayala, 2012*), however, these studies were based on limited sets of genetic markers, which lack the necessary discriminatory power to distinguish between predominantly clonal evolution, with occasional bouts of genetic exchange, and strictly asexual propagation. Genomic-level analyses of *T. brucei* diversity to date have concentrated on *T.b. brucei* and *T.b. rhodesiense* and for *T.b. gambiense* Group 1, include only the genome reference strain (DAL972) (*Goodhead et al., 2013*) or two (*Sistrom et al., 2014*) field isolates. We hereby present a population-level genomic analysis as a means to determine whether this species is truly asexual, when the switch to asexuality arose and to provide insights into the genomic consequences of asexual evolution, including possible compensating strategies for eliminating deleterious mutations.

## Results

The genomes of 75 isolates of *T.b. gambiense* Group 1 (*Supplementary file 1*) were sequenced, including multiple samples from geographically separated disease foci within Guinea (n=37), Côte d'Ivoire (n=36) and Cameroon (n=2) collected over fifty years (1952–2004). For comparative purposes, isolates of *T.b. rhodesiense* (n=4), *T.b. gambiense* Group 2 (n=4) and *T.b. brucei* (n=2) were also sequenced. A total of 230,891 single nucleotide polymorphisms (SNPs) were identified compared to the haploid consensus assembly of the *T.b. brucei* reference genome (*Berriman et al., 2005*). These were evenly distributed over the eleven major chromosomes, covering 85% of the genome (*Figure 1—figure supplement 1*). *T.b. gambiense* Group 1 showed a 5–10 fold lower number of SNPs (11,398) and SNP density compared to the other groups (*Figure 1—source data 1*), despite an over-representation in terms of the number of samples. Phylogenetic network analysis revealed that *T.b. gambiense* Group 1 genotypes showed an extremely low level of intra-group diversity (e.g. the two most distantly related isolates differed only at 435 SNP loci) and formed a monophyletic group (*Figure 1A*). The network features reticulation among non-Group 1 parasites indicating the presence of recombinant genotypes; this stands in contrast to Group 1 parasites and is consistent with an absence, or rarity, of recombination in this group. Network analysis of *T.b. gambiense* Group 1 revealed the population is geographically sub-structured (*Figure 1B*). A significant deviation from Hardy-Weinberg Equilibrium (HWE) was observed at 97.4% of SNP loci (P<10$^{-17}$ at each locus) and this was found to be associated with every sampled genotype being heterozygous at these loci (*Figure 1—source data 2*). To control for geographical and temporal population sub-structure, isolates from three sub-populations were analysed and HWE deviation and heterozygote excess was confirmed (*Figure 1—source data 2*). $F_{IS}$ was calculated for each SNP locus, giving a uni-modal distribution with a median of -1 (*Figure 1—figure supplement 2* and *Figure 1—source data 3*), as would be predicted for a strictly asexual population. Using a genome-wide panel of SNP loci, strong evidence of linkage disequilibrium (LD) was obtained for each chromosome and the whole genome formed a single genetic linkage group (*Figure 1—figure supplement 3*).

Inspection of the SNP distribution across the genome of Group 1 isolates identified multiple long tracts of homozygosity, termed Loss of Heterozygosity (LOH) (*Figure 2—figure supplement 1*). Examination of read depth variation confirmed that this is not due to the loss of part of a homologue and is consistent with mitotic gene conversion; similarly there is no evidence of aneuploidy. In *T.b. gambiense* Group 1, LOH has occurred on every chromosome, with many isolates showing the same LOH patterns (*Figure 2—figure supplement 1*), strongly suggesting that many of these regions arose early in Group 1 evolution. Chromosome 10 displayed the greatest variation in LOH sites, with a total of eighteen different patterns of variation among Group 1 genomes (*Figure 2*). This varied from 2% to 82% of the chromosome length, with LOH occurring predominantly towards one telomere. Distinct LOH patterns were associated with particular phylogenetic lineages, indicating LOH has occurred at various points in Group 1 evolution (*Figure 2—figure supplement 2*).

Taken together, these analyses clearly demonstrate that Group 1 parasites are evolving asexually. A key predicted feature of asexual diploid species is the independent evolution and divergence of alleles on chromosome homologues (*Birky, 1996*), often referred to as the 'Meselson effect' (*Birky, 1996*; *Butlin, 2002*; *Mark Welch and Meselson, 2000*). To test whether this phenomenon occurred in *T.b. gambiense* Group 1, three regions of the genome were chosen where it was possible to manually phase the genomic data, using LOH events to guide the phasing in non-LOH closely related isolates (Materials and methods). For all three regions, a clear sequence of the accumulating mutations could be inferred (*Figure 3*), with each haplotype evolving independently, thus illustrating the Meselson effect (*Judson and Normark, 1996*).

To test whether haplotypes have evolved independently at the whole-chromosomal level, the sequence dataset for all sub-species was computationally phased and haplotypes inferred for each isolate. Excluding LOH regions, phylogenetic trees for each chromosome were generated, revealing that A and B haplotypes separate into distinctive clusters (example in *Figure 4A*) and for every chromosome this pattern was maintained (*Figure 4—figure supplement 1*). Co-phylogenetic analysis revealed congruence between the A and B haplotype trees across different chromosomes (*Figure 4B*; *Figure 4—figure supplements 2* and *3*) illustrating that these haplotypes are evolving in parallel.

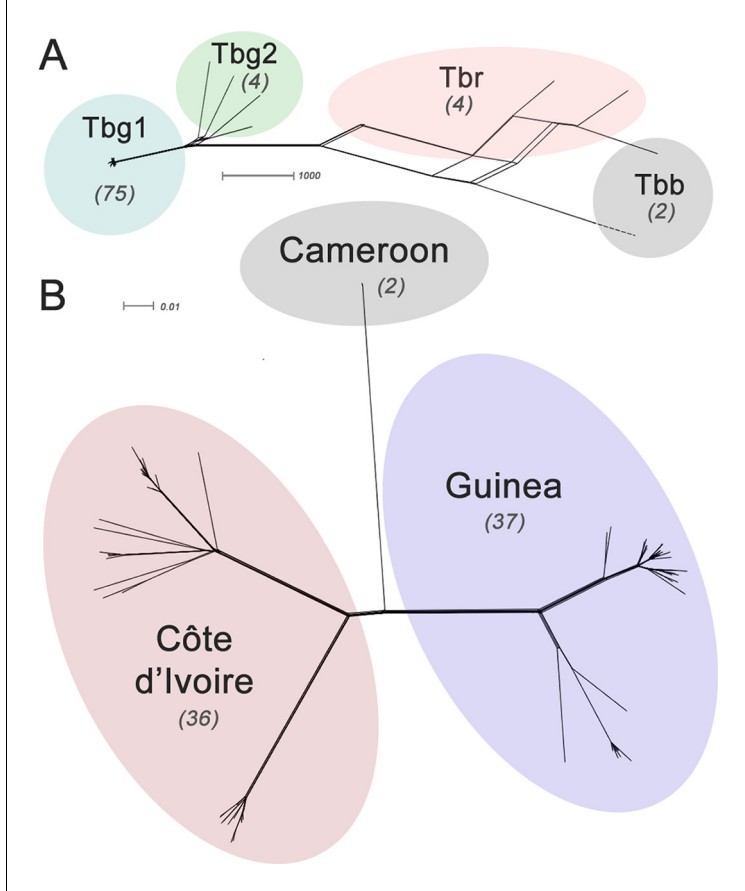

**Figure 1.** Phylogenetic network analysis. SplitsTree phylogenetic networks were constructed using (**A**) each isolate for the collection of *T.b. brucei* (Tbb), *T.b. rhodesiense* (Tbr), *T.b. gambiense* Group 1 (Tbg1) and *T.b. gambiense* Group 2 (Tbg2) and (**B**) for just *T.b. gambiense* Group 1 isolates. The number of samples in each group is indicated in parenthesis.

The following source data and figure supplements are available for figure 1:

**Source data 1.** Number of SNP loci with respect to different sub-species.

**Source data 2.** Testing Hardy-Weinberg Equilibrium (HWE) across the *T.b. gambiense* Group 1 genome.

**Source data 3.** $F_{IS}$ by sub-population.

**Source data 4.** Number and type of *T.b. gambiense* Group 1 SNPs.

**Figure supplement 1.** Genome-wide SNP density map for each sub-species.

**Figure supplement 2.** Weir and Cockerham's $f_{is}$.

**Figure supplement 3.** Genome-wide linkage disequilibrium (LD) among *T.b. gambiense* Group 1 parasites.

---

The time of emergence of *T.b. gambiense* Group 1 was determined. We estimated the genome-wide mutation rate using the number of accumulated mutations (both genome-wide and on Chromosome 9) together with the year of isolation for each isolate, using root-to-tip linear regression (*Drummond et al., 2003*) (*Figure 3—source data 1*). A rate of 1.82 x $10^{-8}$ substitutions per site per year was estimated, similar to that of the COII-NDI 'clock' locus in *T. cruzi* (*Lewis et al., 2011*). Given the observed number of accumulated mutations per isolate, the genome size, and our calculated

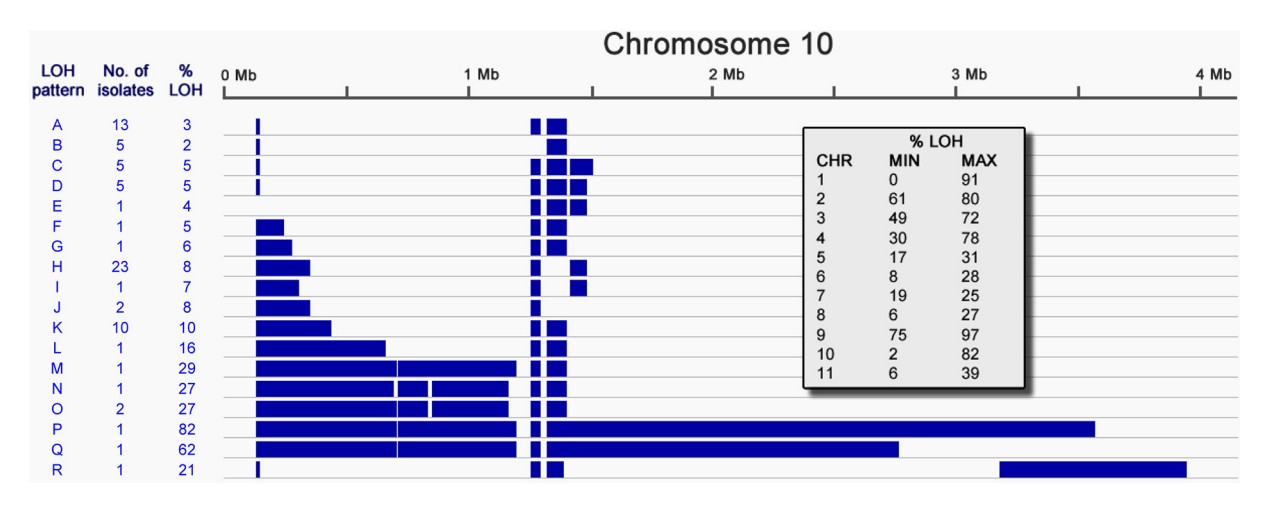

**Figure 2.** Loss of heterozygosity on chromosome 10. Loss of heterozygosity regions (blue) spanning Chromosome 10 show 18 different patterns (A-R). The number of isolates possessing each pattern and the percentage of the chromosome affected are indicated. The table (inset) shows the extent of LOH (min and max) for each chromosome as a percentage of chromosome length.

The following figure supplements are available for figure 2:

**Figure supplement 1.** Loss of heterozygosity across the *T.b. gambiense* Group 1 genome.

**Figure supplement 2.** Loss of heterozygosity across chromosome 10.

mutation rate, the time since the existence of the most recent common ancestor (MRCA) of the Group 1 isolates analysed in the study is estimated to be in excess of one thousand years before present (*Figure 3—source data 1*). Similarly, using the mutation rates for two different *T. cruzi* clock genes (COII-NDI and GPI) (*Lewis et al., 2011*), the date of emergence was estimated to be between 750 and 9,500 years before present (*Figure 3—source data 1*).

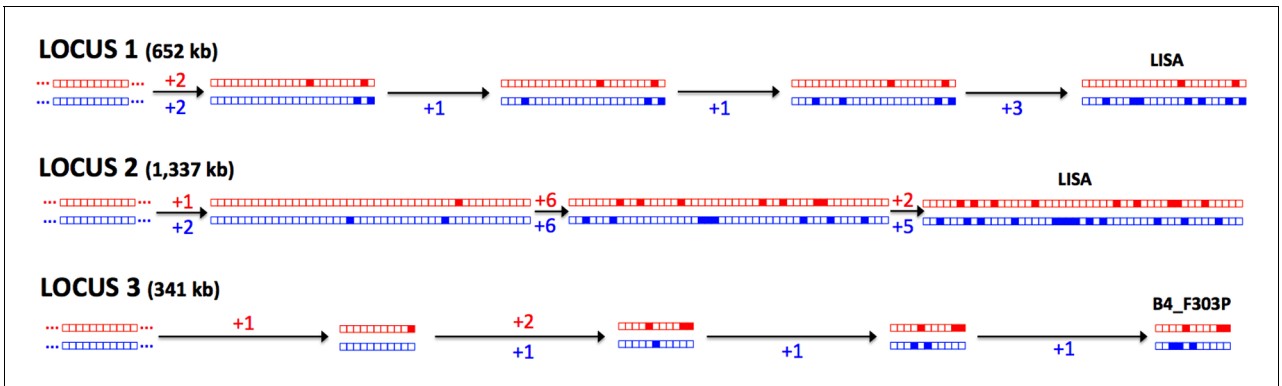

**Figure 3.** The Meselson effect. An accumulation of mutations on separate haplotypes, the 'Meselson Effect', is illustrated using three regions on chromosome 10. For each region (1, 2 and 3) the two haplotypes are shown for a series of isolates with the accumulating mutations (filled boxes) indicated in red or blue for each haplotype. The sequences of accumulating mutations observed in the isolates 'Lisa' and 'B4_F303P' are shown. The number of mutations arising between each ancestral haplotype pair is indicated with two distinct lineages apparent at each locus, illustrating the independent evolution of each haplotype.

The following source data is available for figure 3:

**Source data 1.** Estimated time since the most recent common ancestor.

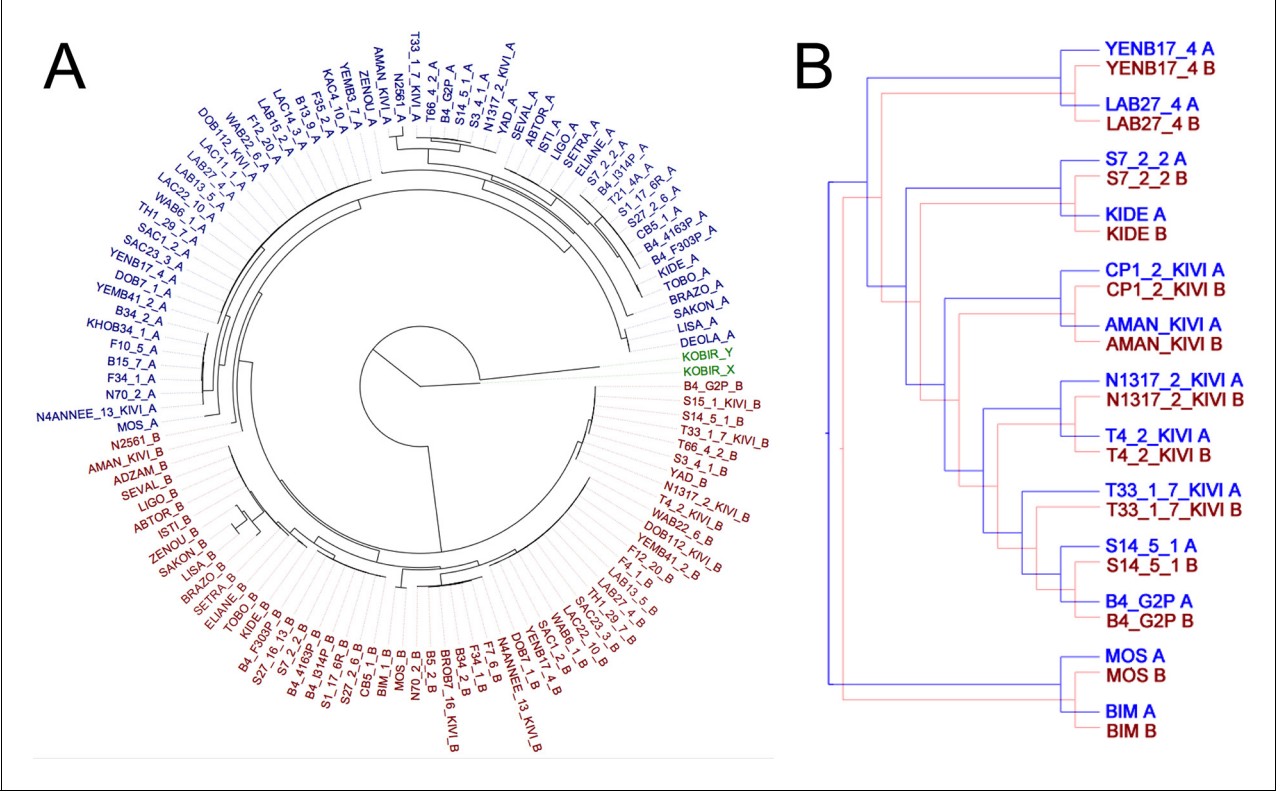

**Figure 4.** Haplotype co-evolution of chromosome 8. (**A**) Phylogenetic tree of phased haplotype sequences of chromosome 8 shows a complete divergence between A (blue) and B (red) haplotypes of *T.b. gambiense* Group 1 (identical genotypes removed). The tree is rooted using a Group 2 isolate (green); (**B**) Co-phylogenetic analysis reveals 100% consensus between the A and B haplotype trees of a subset of *T.b. gambiense* Group 1 isolates.

The following source data and figure supplements are available for figure 4:

**Source data 1.** Co-phylogenetic analysis.

**Figure supplement 1.** Phylogenetic trees of phased data showing 'A' and 'B' haplotypes.

**Figure supplement 2.** Co-phylogenetic analysis.

**Figure supplement 3.** Phylogenetic tree of all *T.b. gambiense* Group 1 isolates.

## Discussion

The theory of clonality in parasitic protozoan populations has been proposed and debated over the last quarter of a century (*Tibayrenc and Ayala, 2002*; *Tibayrenc et al., 1990*). To advance our understanding of clonal evolutional, we have undertaken a whole-genome population-level analysis of *T.b. gambiense* Group 1 focussing on a large number of isolates sampled from two countries, together with a small out-group from a more distant West African country. We provide robust evidence that this important human parasite reproduces exclusively asexually, demonstrating complete genetic linkage across the genome and the absence of allelic segregation. Despite meiosis-specific genes being intact and expressed (*Peacock et al., 2014*), the population genetic data is incompatible with sexual reproduction, self-fertilisation, aneuploidy (*Schurko et al., 2009*), a parasexual cycle (*Ramírez and Llewellyn, 2014*; *Forche et al., 2008*) or the atypical meiosis observed in Rotifers (*Signorovitch et al., 2015*). The barrier to sexual reproduction in *T.b. gambiense* Group 1 remains unclear. The lack of decay of 'meiosis-associated' genes may be explained by a number of non-mutually exclusive hypotheses including the relatively recent emergence of this asexual lineage, such

that insufficient time has elapsed to allow decay. Alternatively, these genes may perform additional roles in processes such as DNA repair or VSG-related recombination.

Our data indicates the parasite population comprises just two independently evolving haplotypes; this remarkable observation confirms the Meselson effect at a whole-genome level for the first time. Despite being predicted for almost twenty years, empirical evidence for this phenomenon has been elusive. The original report of the Meselson effect focused on Bdelliod rotifers (*Welch, 2000*), however this was later shown to instead be due to the entirely different phenomenon of cryptic tetra-ploidy (*Mark Welch et al., 2008*). Subsequent attempts to demonstrate the Meselson effect have been thwarted, such as in the case of the obligately apomictic crustacean, *Daphnia* (*Omilian et al., 2006*). The relatively high rate of mitotic recombination in comparison to the mutation rate results in novel heterozygous sites being eliminated by gene conversion (LOH) events much faster than they are generated and consequently allelic divergence is not observed (*Omilian et al., 2006*). More recently, in the genome of *Timema* stick insects, nuclear alleles have been shown to display a higher level of divergence in asexual rather than sexual species (*Schwander et al., 2011*). In that system, as in *T.b. gambiense* Group 1, the mitotic recombination rate is sufficiently low so as not to obscure the pattern of accumulating mutations, and this has underpinned our ability to detect and confirm the Meselson effect.

The similarity of the genomes studied from different geographical locations, together with a lack of recombination in the evolution of *T.b. gambiense* Group 1, suggests this sub-species emerged from a single progenitor. Each clade represents a separate country and these are partitioned by the earliest branches of the phylogenetic tree, implying early radiation and colonisation. The emergence of *T.b. gambiense* Group 1 within the last 10,000 years coincided with an important period in human history when civilisation and livestock farming were developing in West Africa (*Oliver, 1966*), but whether asexual reproduction was a prerequisite for human infection or occurred subsequently is uncertain. This successful asexual human pathogen contrasts markedly with the virtually extinct sexual *T.b. gambiense* Group 2 parasite that occupied similar biological and geographical niches and this may be an example of the dominance of an asexual lineage over its sexual counterpart (*Charlesworth, 1980*). Another remarkable feature of the *T.b. gambiense* Group 1 genome is the extensive loss of heterozygosity across large regions of each chromosome, which likely arose from gene conversion/mitotic recombination. Gene conversion provides a mechanism for removing a proportion of deleterious mutations in asexual eukaryotes, as hypothesised in other systems (*Tucker et al., 2013*). A fitter allele on one haplotype would be positively selected, resulting in long-range tracts of LOH that encompass the loci under selection together with extensive flanking regions. However, it has been predicted that following an LOH event, individuals will experience a slight decline in long-term fitness due to newfound homozygosity featuring pre-existing sub-optimal alleles, leaving a signature equivalent to inbreeding (*Tucker et al., 2013*) in the progeny. This process has been described as a more powerful evolutionary force than the accumulation of point mutations (*Omilian et al., 2006*) and thus may drive Muller's Ratchet more quickly than *de novo* mutations. This suggests that although LOH may effectively counteract the accumulating mutational load in the short-term, it is unclear whether it can prevent this uniquely well-adapted pathogen from ultimately entering an extinction vortex.

## Materials and methods

### Sample collection

A panel of eighty-five DNA samples was collected, representing *T. brucei* isolates from East and West Africa (*Supplementary file 1*), including *T.b. brucei* (n=2), *T.b. rhodesiense* (n=4), *T.b. gambiense* Group 1 (n=75) and *T.b. gambiense* Group 2 (n=4). The main focus of the study was human-derived *T.b. gambiense* Group 1, with the samples collected from sleeping sickness patients in Guinea (n=37), Côte d'Ivoire (n=36) and Cameroon (n=2). This included collections from Bonon (n=14, collected 2000–2004) in the Côte d'Ivoire and Boffa (n=18, collected 2002) and Dubreka (n=19, collected 1998–2002) in Guinea.

## Illumina sequencing and SNP calling

The *T. brucei* genome is approximately 26 Mb in size and comprises eleven major chromosomes between one and six megabases together with an array of intermediate and mini-chromosomes (*Ogbadoyi et al., 2000*). Illumina paired-end sequencing libraries were prepared from genomic DNA and sequenced by standard procedures on Illumina HiSeq machines, to yield paired sequence reads of 75 bases in length. For each parasite strain, the data yield from the sequencing machines passing the default purity filter was between 7.4 million and 40.4 million read pairs (median of 15.3 million), which corresponds to a nominal genome coverage of between 37.9-fold and 207.1-fold (median of 78.2-fold). For the purpose of calling SNPs, mapping of the paired sequencing reads to the genome reference sequence from GeneDB (*Trypanosoma brucei brucei* TREU927, referred to here as Tb927) was carried out with SMALT (www.sanger.ac.uk/resources/software/smalt/) version 0.7.4 using the following parameters: word length = 13, skip step = 3, maximum insert size = 800, minimum Smith-Waterman score = 60, and with the exhaustive search option enabled. A median fraction of 79.8% of sequencing reads were mapped and a median fraction of 64.9% of sequencing reads were classified as 'proper pairs', i.e. with the two mates of a sequencing read pair mapped within the expected distance and in the correct orientation. Only sequence reads mapped as 'proper pairs' were used for SNP calling, and the first 5 and last 15 nucleotides were clipped from all reads prior to subsequent analysis. Genotypes for every genomic position were determined using SAMtools version 0.1.19 (*Li et al., 2009*) by using the 'samtoolsmpileup' command with minimum baseQ/BAQ ratio of 15 (-Q) followed by SAMtools' 'bcftools view' command with options -c and -g enabled. SNP calls were filtered according to the following criteria: a minimum of 6 high-quality base calls (DP4); a minimum and maximum coverage depth (DP) of 0.25 times and 4 times the median, respectively; a minimum quality score (QUAL) of 22; a minimum mapping quality (MQ) of 22; a minimum second best genotype likelihood value (PL) of 0.25 times the median; a maximum fraction of conflicting base calls for homozygous genotype calls of 10%; and a minimum percentage of 5% for base calls (as a fraction of all base calls for a given genotype) that mapped either to the forward or the reverse strand of the reference sequence. Only loci that passed the quality control criteria for every sample were used for the population analysis. To ensure the SNP-calling parameters used in our analysis did not skew the distribution of variant sites detected (a) within individual genotypes and (b) across the sample collection, SNP-calling was performed using a range of stringencies. While the total number of SNP loci identified per individual and throughout the population varied depending on the stringency of the filter, the allele frequency spectrum remained constant (data not shown). For the analysis presented in the manuscript, our SNP-calling parameters corresponded with a relatively low stringency filter, which was considered capable of detecting a very high proportion of variant sites while ablating the effects of sporadic read errors. Thus, the filter was designed to eliminate artefactual variants in the first instance, although a relatively low number of variant sites may remained undetected due to the necessity for every sample to pass QC at a given locus. The entire SNP dataset is deposited at the TritrypDB pathogen database, which is freely accessible at http://www.tritrypdb.org/tritrypdb/.

## Panels of SNPs for population analysis

A subset of SNP loci was selected where a high-confidence genotype could be identified for every sample in the dataset (*Figure 1—source data 1*). This totalled 230,891 bi-allelic markers, which were used as the basis for the population genomic analyses presented in this study. The number of SNP loci was calculated for each sub-species using two methods: (1) in comparison to the Tb927 reference genome; and (2) in comparison to other members of that sub-species (*Figure 1—source data 1*). A total of 130,180 SNP loci were identified among the seventy-five Group 1 isolates in comparison to the reference *T.b. brucei* genome although only 11,398 of these showed polymorphism within Group 1 itself. In order to facilitate different types of analysis, a series of panels of a sub-set of SNP loci were defined (*Figure 1—source data 4*). For some analyses, Loss of Heterozygosity (LOH) regions of the genome were excluded and therefore a sub-set of SNP loci in non-LOH regions of the genome were identified (*Figure 1—source data 4*; n=5201). In addition, SNP loci in non-LOH regions where the minimum allele frequency was greater than 20% were identified, n=3,549 (*Figure 1—source data 4*), in order to provide sufficient power for testing linkage disequilibrium among the sequenced samples. In order to investigate whether SNP loci where the minimum allele

frequency (MAF) is low correspond to localised areas where recombination events have occurred, the distribution of these loci was visually compared with the distribution of other SNPs in the non-LOH regions of the genome. Similar to the other SNPs, these SNP loci were evenly distributed over the non-LOH regions of the genome, excluding this possibility (data not shown). Finally, a set of SNP loci was identified over non-LOH regions of the genome, excluding fixed heterozygous loci, which was polymorphic only among *T.b. gambiense* Group 1 isolates. These correspond to 'derived' alleles, which arose since the most recent common ancestor of the Group 1 isolates studied (*Figure 1—source data 4*, 'Tbg1 derived'). A set of SNPs loci polymorphic both within and outside the Group 1 population was also defined (*Figure 1—source data 4*, 'Tbg1 ancient').

## Phylogenetic networks and trees

Phylogenetic networks were constructed using the Split Decomposition method of SplitsTree4 (*Huson and Bryant, 2006*): *Figure 1A* shows the reconstruction using all the isolates sequenced in this study (n=85) and the full panel of SNPs; *Figure 1B* shows relationships among the Group 1 isolates (n=75) using the SNP panel corresponding to derived alleles in non-LOH areas (*Figure 1—source data 4*). The virtually non-reticulated topology of the network presented in *Figure 1B* supports our finding of strict asexuality in the Group 1 population and it is therefore appropriate to utilise a classical phylogenetic tree approach for the analysis of this sub-species. Maximum Likelihood phylogenetic trees were constructed using RAxML (*Stamatakis, 2014*) using a generalised time-reversible model of sequence evolution. Confidence in individual branching relationships was assessed using 100 bootstrap pseudo-replicates and trees visualised using FigTree 1.4 (tree.bio.ed.ac.uk).

## Linkage disequilibrium and Hardy-Weinberg equilibrium

The Haploview software package (*Barrett, 2009*) was used to investigate LD across each chromosome using unphased data and to calculate the normalised measure of allelic association, D' (*Daly et al., 2001*). Linkage blocks were defined using the method of *Gabriel et al. (2002)* with a block being created if 95% of informative comparisons between SNP loci were found to possess 'strong LD'. Strong LD was defined as D'=1 and LOD score $\geq$2. Haploview was also used to calculate the probability that any observed deviation from HWE could be explained by chance using a $\chi^2$ test. This was performed for SNP loci across the *T.b. gambiense* Group 1 genome utilising the set of 'Tbg1 ancient' SNP loci (*Figure 1—source data 4*). Statistically significant loci at P<0.001 were additionally tested to determine whether deviation from HWE was associated with a heterozygote excess (*Figure 1—source data 2*), by comparing the predicted with the observed heterozygote frequency. Overall, a high proportion of SNP loci (97.4%) showed a statistically significant departure from HWE (P<$10^{-17}$) and strikingly, all of these loci showed an excess of heterozygotes (*Figure 1—source data 2*). Along with the entire set of *T.b. gambiense* Group 1 isolates, a separate HWE analysis was performed on three spatio-temporally defined sub-populations (Bonon, Boffa and Dubreka), thus accounting for any geographical and temporal population sub-structure. A series of further genetic tests was performed using Fstat version 2.9 (*Goudet, 1995*). $F_{IS}$ was calculated across the *T.b. gambiense* Group 1 genome, again utilising the set of 'ancient' SNP loci (*Figure 1—figure supplement 2*). A median figure of -1 was calculated for the entire set of *T.b. gambiense* Group 1 isolates and for each of the Bonon, Boffa and Dubreka sub-populations, indicating strict asexuality. This was supported by permutation testing (n iterations = 30,000), which indicated that $F_{IS}$ was lower than expected (*Figure 1—source data 3*). Similarly, using the Bonon, Boffa and Dubreka sub-populations, Weir & Cockerham's $f_{is}$ was also calculated using Fstat. This had a uni-modal distribution with a median of -1 ($f_{is}$ = -1 at 91.4% of loci), indicating strict asexuality.

## 'Loss of heterozygosity' analysis

To assess the distribution of heterozygous sites across the genome, the density of these sites was calculated in 10 kb segments for every isolate. These density figures were used to determine whether each 10 kb segment could be considered a candidate area for long-range LOH. LOH blocks were defined using a custom Perl script to perform Interval Analysis using the following criteria: max number of heterozygous sites allowed per block = 0, minimum number of contiguous blocks = 6, maximum gap size in a contiguous block = 2, max number of heterozygous sites allowed within gap

= 2. LOH block data was converted for viewing in the Integrative Genome Viewer (IGV) (*Thorvaldsdottir et al., 2013*). In order to determine whether genomic structural variation could explain observed LOH, we performed a systematic copy number variation (CNV) analysis across the genome using CNVnator (*Mills et al., 2011*). This reveals that there was no loss or gain of chromosomal material associated with LOH segments (data not shown).

## Manual phasing, computational phasing and co-phylogenetic analysis

In order to validate the computational phasing and investigate the relationship between haplotypes, three loci were selected where recent LOH events had occurred independently on chromosome 10 in different isolates. Such independent LOH events may be identified by examining patterns of LOH in comparison to the phylogenetic tree (*Figure 2—figure supplement 2*). For locus 1, for example, an LOH region recently arose independently in B7_2 and DEOLA and can be observed over approximately 650 kb of chromosome 10 (*Figure 2—figure supplement 2*). Examination of the 506 SNP loci identified in this region indicated that two ancestral haplotypes existed. The nearest neighbours of these isolates (B7_2 and DEOLA) on the phylogenetic tree, CP1_2_KIVI and LISA, respectively, did not possess LOH in this region and therefore the sequences of B7_2 and DEOLA could be used as guides to allow CP1_2_KIVI and LISA to be confidently phased. Since the divergence of CP_1_2_KIVI with LISA, at this locus 15 mutations arose in the former isolate and 9 mutations in the latter and this is illustrated for LISA in *Figure 3*. Again, because of the existence of only two haplotypes, more distant isolates could also be phased in this manner, which allowed intermediate haplotypes to be inferred and the accumulating sequence of Meselson mutations to be determined.

To permit a genome-wide analysis, computational phasing of the diploid genotypic data was performed using the segmented haplotype estimation and imputation tool SHAPEIT2 (*Delaneau et al., 2012*; *Delaneau et al., 2013*). The default parameters were used together with an adjusted window size of 0.5 Mb and a recombination rate of 0.0003 (*MacLeod et al., 2005*). The accuracy of the computational phasing for each isolate was assessed in comparison to a large LOH region on one isolate (DEOLA at the 650 kb Locus 1 on Chromosome 10). LOH in this region provided a set of 'gold standard' phasing information, which was used to check the phasing of all isolates, except the five which shared LOH in this region. A switch error rate (*Lin et al., 2004*) of between 4% and 12% was observed (mean 9.3%) across the 69 isolates, validating the results of the computational phasing.

Phased sequence data from all isolates in the collection was used to create a separate Maximum Likelihood phylogenetic tree for each chromosome with RAxML (*Stamatakis, 2014*) (*Figure 4—figure supplement 1*). For *T.b. gambiense* Group 1 isolates, co-phylogeny of the phased haplotypes (A vs B) was then assessed for each chromosome in turn using Jane (*Conow et al., 2010*). Tree topologies were resolved using a Genetic Algorithm for co-phylogeny reconstruction with the default cost model. To assess whether the trees were more similar than would be expected by chance, 1,000 simulations were carried out using each of: (a) a random tip-mapping method; and (b) a random tree method (beta = -1). For every chromosome, A and B haplotype trees were significantly similar to each other (*Figure 4—source data 1*, P < 0.001). In order to illustrate this similarity between A and B haplotype trees, a set of 27 isolates was selected which could be resolved with 100% bootstrap support from a phylogenetic tree constructed using the whole-genome dataset (*Figure 4—figure supplement 3*). The three chromosomes with the largest number of SNPs among *T.b. gambiense* Group 1 (8, 10 and 11) were then selected, as these were the most informative. Isolates which could be resolved by the most highly supported nodes in trees representing the A and B haplotypes were subsequently selected and this corresponded to 16, 18 and 13 isolates for chromosomes 8, 10 and 11 respectively. In each case, the A and B haplotype sub-trees showed identical topology (*Figure 4—figure supplement 2*), illustrating the co-evolution of partner haplotypes.

## Dating the emergence of *T.b. gambiense* Group 1

The date of emergence of *T.b. gambiense* Group 1 was calculated by combining the mutational rate with the estimated number of mutations arising since the most recent common ancestor of our collection of Group 1 isolates. The mutation rate was calculated using two approaches, the first using

all seventy-five isolates, the second using a subset of samples representing a discrete lineage isolated in the Côte d'Ivoire (*Figure 3—source data 1*). Both methods gave rise to a similar mutation rate of approximately 2 x 10$^{-8}$ mutations per base per year.

Two approaches were used to assess the number of accumulated mutations present in the genome of each isolate. In the first, only the non-LOH portion of the genome was utilised to avoid the risk of mutations being 'erased' by way of LOH/gene conversion events. The number of SNP loci where derived alleles were present in the genome was counted. 388 such loci (+/- 12) were identified on both homologues over 8.47 Mb of the non-LOH portion of the genome. The second involved focusing on Chromosome 9, where a large region of LOH is found over much of the chromosome in every Group 1 isolate. Any mutations occurring since the existence of the most recent common ancestor of all the isolates analysed can easily be identified as they manifest themselves as heterozygous loci; this avoids the issue of identifying fixed heterozygous loci, which represent uninformative pre-existing mutations rather than accumulated mutations. Similar results were achieved with both methods, providing us with estimates of the time to most recent common ancestor (TMRCA) of approximately 1,000 to 1,500 years before present (*Figure 3—source data 1*). The mutation rates of two loci in *T. cruzi*, for which a 95% confidence interval was available, were also utilised to date the emergence of Group 1 parasites. These provided a confidence interval of approximately 750 to 9,500 years before present (*Figure 3—source data 1*).

## Acknowledgements
We thank Lucio Marcello for providing technical assistance in the LOH analysis.

## Additional information

### Funding

| Funder | Grant reference number | Author |
|---|---|---|
| Wellcome Trust | 095201/Z/10/Z | William Weir<br>Paul Capewell<br>Caroline Clucas<br>Andrew Pountain<br>Pieter Steketee<br>Nicola Veitch<br>Anneli Cooper<br>Annette MacLeod |
| Wellcome Trust | 085349 | William Weir<br>Paul Capewell<br>Caroline Clucas<br>Andrew Pountain<br>Pieter Steketee<br>Nicola Veitch<br>Anneli Cooper<br>Annette MacLeod |
| Wellcome Trust | 098051 | Bernardo Foth<br>Matt Berriman |

The funders had no role in study design, data collection and interpretation, or the decision to submit the work for publication.

### Author contributions
WW, performed the genomic and genetic analysis and wrote the paper; PC, performed the computational phasing; BF, performed the SNP calling; CC, NV, prepared parasite material and purified DNA; AP, was involved in LOH analysis; PS, was involved in LOH analysis; MK, JK, BB, Co-ordinated and performed field sampling and prepared parasite DNA; TDM, provided expert support in analysing an interpreting the data; MC, Co-ordinated and performed field sampling and prepared parasite DNA ; AC, was involved in LOH analysis ; AT, wrote the paper; VJ, Co-ordinated and performed field sampling and prepared parasite DNA ; MB, supervised the sequencing; AMACL, Conceived the study and wrote the paper

**Author ORCIDs**

Thierry De Meeûs, http://orcid.org/0000-0001-8807-241X

Matt Berriman, http://orcid.org/0000-0002-9581-0377

## Ethics

Human subjects: Overall ethical approval for the study was granted by the University of Glasgow College of Medical, Veterinary Life Sciences Ethics Committee (project number 200120043). Guinean parasite strains were collected with local ethical approval within the framework of medical surveys conducted by the national Human African Trypanosomiasis (HAT) control program (NCP) according to the national HAT diagnostic procedures of the Republic of Guinea Ministry of Health. No samples other than those collected for routine screening and diagnostic procedures were collected and all human samples were anonymised. All participants were orally informed of the objective of the study in their own language by an NCP health officer. This study is part of a larger project for which approval was obtained from the WHO Research Ethics Review Committee (RPC222) and Institut de Recherche pour le Développement (Comité Consultatif de Déontologie et d'Ethique) ethical committee. Ivory Coast parasite strains were collected during medical surveys conducted by the Ivory Coast NCP in agreement with the National Ministry of Health and in collaboration with the IRD, according to WHO and Ivory Coast NCP recommendations. Patients who gave their consent after explanation of the objective and rationale of the study were used in this work. In both countries, all confirmed cases were offered treatment.

## Additional files

### Supplementary files

• Supplementary file 1. Isolates used in this study. For each isolate, the year of isolation, host, country and location are given along with the results of the BIIT test (Blood Incubation Infectivity Test), which determines human infectivity. The presence/absence of *TgSGP*, the *T.b. gambiense* Group 1 human serum resistance gene and *SRA*, the *T.b. rhodesiense* human serum resistance gene are indicated. The majority of samples in this study were *T.b. gambiense* Group 1, details of which have been previously published (*Heitman, 2006*; *Thorvaldsdottir et al., 2013*).

### Major datasets

The following datasets were generated:

| Author(s) | Year | Dataset title | Dataset URL | Database, license, and accessibility information |
|---|---|---|---|---|
| Weir W, Foth B, Clucas C, Pountain A, Steketee P, Veitch N, Koffi M, De Meeûs T, Kaboré J, Camara M, Cooper A, Tait A, Jamonneau V, Bucheton B, Berriman M, MacLeod A | 2016 | Tritrypdb SNP collection | http://tritrypdb.org/tritrypdb/ | NA |

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
