## [Decision Letter]

Thank you for submitting your work entitled "Population genomics reveals the origin and asexual evolution of human infective trypanosomes" for consideration by *eLife*. Your article has been reviewed by three peer reviewers, and the evaluation has been overseen by a Reviewing Editor, Dominique Soldait-Favre and Diethard Tautz as the Senior Editor.

The reviewers have discussed the reviews with one another and the Reviewing editor has drafted this decision to help you prepare a revised submission.

Summary:

The study reports a population genomics approach applied to the examination of the irreversible accumulation of mutations in the asexually reproducing diploid *Trypanosoma brucei gambiense*. The authors provide persuasive evidence for the absence of sex and recombination during a substantial period of the evolution of *T. brucei gambiense*. This manuscript features a spectacular case of the Meselson effect at a genome-wide level. It clearly shows divergence between homologous chromosomes and loss of heterozygosity that the authors interpret as a compensatory mechanism for counteracting deleterious mutations.

As well as being important for understanding the evolution of this pathogen, it may become a textbook example of the population genetics of an asexual diploid, thus advancing the field of evolutionary biology.

Essential revisions:

The analysis is very comprehensive and accessible to non-specialists and the reviewers mainly identified some minor issues. However the results presented here are far from being expected, since until now, this phenomenon has not been observed in other asexual organisms. In consequence the authors should discuss much more extensively the Meselson effect story taking the following points in consideration:

The authors somewhat underplay the surprise of this, characterizing the Meselson effect as "predicted". I'd strongly suggest a slightly fuller account of the history of work on the Meselson effect to provide clearer context of the importance of the present paper. Although the Meselson effect was indeed predicted at one time (Birky, 1996; Butlin, 2002; Welch and Meselson, 200; Judson and Normark, 1996), empirically it has been elusive. The original empirical report of the Meselson effect in bdelloid rotifers (Welch and Meselson, 2000) was later shown to have been due to a different phenomenon entirely: cryptic tetraploidy (Mark Welch, D. B., J. L. Mark Welch, and M. Meselson. 2008. Evidence for degenerate tetraploidy in bdelloid rotifers. Proc. Natl. Acad. Sci. USA 105:5145-5149). The Meselson effect was also found to be absent in obligately apomictic Daphnia, due to the high rate of mitotic recombination in comparison to the mutation rate (Omilian, A., M. Cristescu, J. Dudycha, and M. Lynch. 2006. Ameiotic recombination in asexual lineages of Daphnia. Proc. Natl. Acad. Sci. USA 103:18638-18643). The Meselson effect appeared to require an unrealistically high ratio of mutation to mitotic recombination. The present manuscript is the most thorough and persuasive of the trickle of recent studies indicating that the Meselson effect can in fact happen, in lineages with a sufficiently low rate of mitotic recombination (cf. Schwander, T., L. Henry, and B. J. Crespi. 2011. Molecular Evidence for Ancient Asexuality in Timema Stick Insects. Curr. Biol. 21:1129-1134.) Indeed the swathes of Loss-of-Heterozygosity give a vivid sense of the interplay between recombination and mutation, in a lineage in which recombination is rare enough not to have erased the whole record.

Remarks:

LOH, generated by gene conversion is hypothesized to be a mechanism for removing deleterious alleles. This seems plausible. However to determine that LOH events are selected rather than neutral, a more comprehensive analysis of the distribution, size, gene content and rate of accumulation across time would be informative. For example, I wonder if it is possible to demonstrate that LOH specifically occurs in regions containing phylogenetically conserved genes that are intolerant to mutation accumulation. This is not critical for the current report, but would be well worth following up.

The fine scale congruence between the phylogenies of the A and B genomes, documented in Figure 4, is impressive and goes a long way towards demonstrating that the observed heterozygosity represents evolution within the asexual lineage and is not inherited from, say, a hybrid ancestor. (Initially I was puzzled that it looked like there was incongruence between the chromosomes but closer inspection shows that there is no such incongruence, just sporadic omission of isolates that failed to meet the 1000SNP/chromosome cutoff.)

Author contributions – one author is listed who "commented on the analysis and manuscript". Acknowledgements would be more appropriate here unless a more substantial contribution can be provided.

---

## [Author Response]

*Essential revisions: The analysis is very comprehensive and accessible to non-specialists and the reviewers mainly identified some minor issues. However the results presented here are far from being expected, since until now, this phenomenon has not been observed in other asexual organisms. In consequence the authors should discuss much more extensively the Meselson effect story taking the following points in consideration:*

*The authors somewhat underplay the surprise of this, characterizing the Meselson effect as "predicted". I'd strongly suggest a slightly fuller account of the history of work on the Meselson effect to provide clearer context of the importance of the present paper. Although the Meselson effect was indeed predicted at one time (Birky, 1996; Butlin, 2002; Welch and Meselson, 200; Judson and Normark, 1996), empirically it has been elusive. The original empirical report of the Meselson effect in bdelloid rotifers (Welch and Meselson, 2000) was later shown to have been due to a different phenomenon entirely: cryptic tetraploidy (Mark Welch, D. B., J. L. Mark Welch, and M. Meselson. 2008. Evidence for degenerate tetraploidy in bdelloid rotifers. Proc. Natl. Acad. Sci. USA 105:5145-5149). The Meselson effect was also found to be absent in obligately apomictic Daphnia, due to the high rate of mitotic recombination in comparison to the mutation rate (Omilian, A., M. Cristescu, J. Dudycha, and M. Lynch. 2006. Ameiotic recombination in asexual lineages of Daphnia. Proc. Natl. Acad. Sci. USA 103:18638-18643). The Meselson effect appeared to require an unrealistically high ratio of mutation to mitotic recombination. The present manuscript is the most thorough and persuasive of the trickle of recent studies indicating that the Meselson effect can in fact happen, in lineages with a sufficiently low rate of mitotic recombination (cf. Schwander, T., L. Henry, and B. J. Crespi. 2011. Molecular Evidence for Ancient Asexuality in Timema Stick Insects. Curr. Biol. 21:1129-1134.) Indeed the swathes of Loss-of-Heterozygosity give a vivid sense of the interplay between recombination and mutation, in a lineage in which recombination is rare enough not to have erased the whole record.*

We now discuss the history of the Meselson as requested (Discussion):

“Our data indicates the parasite population comprises just two independently evolving haplotypes; this remarkable observation confirms the Meselson effect at a whole-genome level for the first time. […] In that system, as in *T.b. gambiense* Group 1, the mitotic recombination rate is sufficiently low so as not to obscure the pattern of accumulating mutations, and this has underpinned our ability to detect and confirm the Meselson effect.”

*Remarks: LOH, generated by gene conversion is hypothesized to be a mechanism for removing deleterious alleles. This seems plausible. However to determine that LOH events are selected rather than neutral, a more comprehensive analysis of the distribution, size, gene content and rate of accumulation across time would be informative. For example, I wonder if it is possible to demonstrate that LOH specifically occurs in regions containing phylogenetically conserved genes that are intolerant to mutation accumulation. This is not critical for the current report, but would be well worth following up.*

We thank the editor/reviewers for this comment. We are currently performing further analysis on the LOH regions of the genome to investigate this issue and we hope to publish the results in the early New Year.

*The fine scale congruence between the phylogenies of the A and B genomes, documented in Figure 4, is impressive and goes a long way towards demonstrating that the observed heterozygosity represents evolution within the asexual lineage and is not inherited from, say, a hybrid ancestor. (Initially I was puzzled that it looked like there was incongruence between the chromosomes but closer inspection shows that there is no such incongruence, just sporadic omission of isolates that failed to meet the 1000SNP/chromosome cutoff.)*

We again thank the editor/reviewers for this comment; we are very pleased that the A/B genome phylogeny comparison provides such clear evidence for haplotype co-evolution.

Author contributions – one author is listed who "commented on the analysis and manuscript". Acknowledgements would be more appropriate here unless a more substantial contribution can be provided.

This author’s contribution is more substantial that indicated – he provided expert support on applying classical population genetic analysis to genomic- scale data and provided extensive input on the direction of the manuscript, which focused our analysis on pursuing the Meselson effect.